# *MCD* Inhibits Lipid Deposition in Goat Intramuscular Preadipocytes

**DOI:** 10.3390/genes14020440

**Published:** 2023-02-08

**Authors:** Changheng Yang, Qi Li, Yaqiu Lin, Yong Wang, Hengbo Shi, Lian Huang, Wangsheng Zhao, Hua Xiang, Jiangjiang Zhu

**Affiliations:** 1Qinghai-Tibetan Plateau Animal Genetic Resource Reservation and Utilization Key Laboratory of Sichuan Province, Southwest Minzu University, Chengdu 610041, China; 2Key Laboratory of Qinghai-Tibetan Plateau Animal Genetic Resource Reservation and Utilization (Southwest Minzu University), Ministry of Education, Chengdu 610041, China; 3College of Animal Science, Zhejiang University, Hangzhou 310058, China; 4School of Life Sciences and Engineering, Southwest University of Science and Technology, Mianyang 621010, China

**Keywords:** goat, malonyl-CoA decarboxylase (*MCD*), intramuscular preadipocyte, intramuscular fat (IMF)

## Abstract

Malonyl-CoA decarboxylase (*MCD*) is a major regulator of fatty acid oxidation catalyzing the decarboxylation of malonyl coenzyme A (malonyl-CoA). Although its involvement in human diseases has been well studied, its role in intramuscular fat (IMF) deposition remains unknown. In this present study, 1726 bp of *MCD* cDNA was cloned (OM937122) from goat liver, including 5′UTR of 27 bp, 3′UTR of 199 bp, and CDS of 1500 bp, encoding 499 amino acids. In this present study, although the overexpression of *MCD* increased the mRNA expression of *FASN* and *DGAT2*, the expression of *ATGL* and *ACOX1* was also activated significantly and resulted in a decrease in cellular lipid deposition in goat intramuscular preadipocytes. Meanwhile, the silencing of *MCD* increased the cellular lipid deposition and was accompanied by the expression activation of *DGAT2* and the expression suppression of *ATGL* and *HSL*, despite the expression suppression of genes related to fatty acid synthesis, including *ACC* and *FASN*. However, the expression of *DGAT1* was not affected significantly (*p* > 0.05) by the expression alteration of *MCD* in this present study. Furthermore, 2025 bp of *MCD* promoter was obtained and predicted to be regulated by C/EBPα, SP1, SREBP1, and PPARG. In summary, although different pathways may respond to the expression alteration of *MCD*, the expression of *MCD* was negatively correlated with the cellular lipid deposition in goat intramuscular preadipocytes. These data may be beneficial for enhancing our understanding of the regulation of IMF deposition in goats.

## 1. Introduction

Goat meat is increasingly popular because of its high protein, high calcium, and low cholesterol. Thus, the quality of goat meat is in higher demand. Intramuscular fat (IMF) content affects meat tenderness, flavor, etc. Increasing IMF is an important way to improve goat meat quality [1,2].

Many adipose-related genes and substances are known to affect lipid deposition [3]. Malonyl coenzyme A (malonyl-CoA) has become a central mediator of many important cellular processes, not only as a substrate for fatty acid synthesis catalyzed by fatty acid synthase (FASN) but also as a potent endogenous inhibitor of carnitine palmitoyltransferase 1 (CPT1) to inhibit fatty acid β-oxidation [4,5]. Malonyl-CoA is produced by acetyl-CoA carboxylase (ACC) and decarboxylated to acetyl-CoA via malonyl-CoA decarboxylase (MCD) [6].

Malonyl-CoA decarboxylase, which catalyzes the decarboxylation of malonyl-CoA, was first characterized from the goose uropygial gland [7] and is highly expressed in rat liver, heart, and adipose tissue [8,9]. Although the intracellular localization of MCD is unclear, some studies have shown that MCD is present in mitochondria, peroxisomes, or cytosol [10,11] and involves fatty acid synthesis and oxidation regulation. MCD deficiency in humans resulted in fatty acid metabolism errors and malonic aciduria (Human MLYCD deficiency (OMIM 248,360)) [12,13,14]. In contrast, overexpression of *MCD* in the liver reversed high-fat-diet-induced hepatic and muscle insulin resistance in rats [15]. In addition, MCD plays an important role in fatty acid metabolism during increased cardiac workload, hypertrophy, diabetes [16], and ischemia–reperfusion injury [17,18]. MCD has also been confirmed as a potential cancer treatment target, including breast cancer [19] and lung cancer [20].

Although current studies suggest that MCD is an essential regulator of lipid metabolism in humans or rats, little is known about its role in ruminants. In addition, no studies on the MCD promoter have been reported. Our hypothesis was that MCD played an important role in regulating intramuscular fat deposition. In this present study, we elucidated the function of MCD in regulating cellular lipid metabolism using the treatment of *MCD* knockdown and overexpression in goats. These data may provide important evidence for the role of MCD in intramuscular fat deposition.

## 2. Materials and Methods

### 2.1. Animals, Cell Isolation, and Culture

Two two-day-old male goats were purchased from Sichuan Jianyang Dageda Aminal Husbandry Co., Ltd. (Jianyang, China). The animal studies were approved by the Southwest Minzu University Animal Care and Use Committee. Goat intramuscular precursor adipocytes were isolated as previously described [21]. Briefly, the goats were washed and disinfected after execution, and the longissimus dorsi tissue was rapidly collected under aseptic conditions. Next, it was washed three times with PBS before being cut up with scissors and mixed in equal amounts. The digestion was performed by adding appropriate amounts of collagenase type (Sigma, St. Louis, MO, USA). After digestion for 1.5 h, the supernatant was collected by centrifugation at 2000 r/min for 5 min and filtered through a 75 μm filter. The supernatant was removed by centrifugation, red blood cell (RBC) lysate (Boster, CA, USA) was added, and the supernatant was again removed by centrifugation. Cells were resuspended with DMEM/F12 (Gibco, Beijing, China) containing 10% fetal bovine serum (FBS), and culture was started. Cells were cultured for 2 h, and then the fluid was changed.

### 2.2. Induction of Cell Differentiation

The third-generation cells were treated with 50 μmol/L oleic acid (Sigma, St. Louis, MO, USA) and harvested on days 0, 1, 3, and 6. These samples were used to explore the lipid deposition in goat intramuscular preadipocytes during differentiation and the mRNA expression levels of *MCD*.

### 2.3. Gene Cloning and Bioinformatic Analysis

Goat liver tissue preserved in the laboratory was removed from liquid nitrogen. After grinding with liquid nitrogen, 1 mL Trizol (TaKaRa, Dalian, China) was added for RNA extraction. cDNA was obtained according to the reverse transcription kit’s (Takara, Otsu, Japan) instructions, and the primers were designed according to the predicted sequence of the goat *MCD* gene in GenBank (XM_018061772) using Primer Premier 5.0 software (forward primer: TTCCCCAGGCAGCTGTCGC, reverse primer: CCGGCGGTAATGCGCTTTC). PCR was performed using the touchdown PCR method. PCR reaction system: PrimeSTAR^®^ Max DNA Ploymerase (Takara, Dalian, China) 12.5 μL, cDNA 1 μL, upstream primer 1 μL, downstream primer 1 μL, and ddH_2_O 8.5 μL. The procedure is as follows: 98 °C pre-denaturation for 3 min; 98 °C denaturation for 10 s, 68 °C annealing for 30 s, 72 °C extension for 90 s, each cycle minus 1 °C, 13 cycles; 98 °C denaturation for 10 s, 55 °C annealing for 30 s, 72 °C extension for 90 s, extension at 72 °C for 5 min, storage at 4 °C. Then, PCR product was added to 2 μL 2x Taq 70 °C extension for 30 min. The PCR products were ligated with pMD^®^ 19-T (Takara, Otsu, Japan) vector and sequenced by Beijing Qingke Biotechnology Co. We used Blastn (NCBI) for homology analysis and MEGA5.0 software for constructing phylogenetic trees.

### 2.4. Construction of pcDNA3.1-MCD Plasmid and Synthesis of siRNA

Based on the *MCD* gene sequence obtained from the previous cloning, subclonal primers were designed (forward primer: CCCAAGCTTGCCACCATGGATTACAAGGATGACGACGATAAGCGAGTTCTCGGGCCAAGC, reverse primer: CGCGGATCCCTAGAGTTTGCTGTTCTTCTG). The upstream and downstream primers were added to the H*ind* III and B*amH* I sites (underlined), respectively, and synthesized by Shanghai Bioengineering Co. PCR amplification was performed using pMD19-T-MCD plasmid as a template. PCR reaction system: LA Taq enzyme (Takara, Dalian, China) 12.5 μL, pMD19-T-MCD plasmid 1 μL, upstream primer 1 μL, downstream primer 1 μL, and ddH_2_O 8.5 μL. The procedure is as follows: 94 °C pre-denaturation for 4 min; 98 °C denaturation for 15 s, 64 °C annealing for 30 s, 72 °C extension for 70 s, each cycle minus 1 °C, 13 cycles; 98 °C denaturation for 15 s, 51 °C annealing for 30 s, 72 °C extension for 70 s, extension at 72 °C for 5 min, and storage at 12 °C. PCR products and pcDNA3.1 plasmids were double-digested. Digestion reaction system: pcDNA3.1 plasmids or PCR products 1 ug, K buffer 2 μL, H*ind* III 1 μL, B*amH* I (Takara, Dalian, China) 1 μL, and supplemented with ddH_2_O to 20 μL. The double-enzyme digestion reaction system was placed in a 37 °C water bath for 12 h. The digested products were then purified, and the gene fragment and pcDNA3.1 vector were ligated using T4 ligase (Takara, Dalian, China) in a metal bath at 16 °C for 12 h. Finally, the recombinant plasmids were double-digested and sequenced for identification. The siRNA was synthesized by GenePharma (Shanghai, China). The siRNA-MCD sequence is as follows, sense: 5′-GCAGCAUCCAGACAAUCAUTT-3′, anti-sense: 5′-AUGAUUGUCUGGAUGCUGCTT-3′. Negative control siRNA is as follows, sense: 5′-UUCUCCGAACGUGUCACGUTT-3′, anti-sense: 5′-ACGUGACACGUUCGGAGAATT-3′.

### 2.5. Cell Transfection

When the third-generation cells grew to cover 70–80% of the bottom, the cells were transfected with pcDNA3.1-MCD (OMCD), pcDNA3.1 (ONC), siRNA-MCD (siMCD), and negative control siRNA (siNC), respectively, according to the instructions of Lipofetamine 3000 Transfection Kit (Invitrogen, Carlsbad, CA, USA). In this study, transfer into pcDNA3.1-MCD plasmid (OMCD) or siRNA-MCD (siMCD) was used as the experimental group, and transfer into pcDNA3.1 plasmid (ONC) or negative control siRNA (siNC) was used as the control group. The cells were first starved, the medium was discarded, washed three times with pre-cooled PBS, 900 μL Opti medium was added to each well of the 6-well plate, and the cells were placed in a constant temperature incubator at 37 °C for 4 h. After the starvation treatment, premixes were prepared for transfection. Premix preparation for overexpression experiment: Premix A was prepared by adding 3 μL lip3000 to 50 μL Opti medium. Premix B was prepared with 50 μL Opti medium, 2.5 μL P3000, and 1 μg plasmid (pcDNA3.1-MCD or pcDNA3.1). Premix A and Premix B were mixed thoroughly and let to stand at room temperature for 20 min. Premix preparation for interference experiment: 100 μL Opti medium, 6 μL lip3000, and 4 μL siRNA were mixed well and allowed to stand at room temperature for 10 min. After the premix was finished standing, it was suspended and added dropwise to complete the cell transfection for overexpression and interference experiments. After 6 h of transfection, cells were incubated with a final concentration of 50 μmol/L oleic acid (Sigma) induction solution. Following 48 h of culture, cells were harvested.

### 2.6. Extraction of RNA and Quantitative Real-Time PCR (RT-qPCR)

Extraction of total RNA from cells was completed using Trizol reagent (TaKaRa, Dalian, China). The concentration and integrity of total RNA were determined using a Nanodrop 2000 (Thermo Fisher Scientific, Beijing, China) and 2% agarose gel electrophoresis, respectively. Then, 1 μg of RNA was taken to synthesize cDNA by Revert Aid First Strand cDNA Synthesis Kit (Thermo Fisher Scientific). RT-qPCR primers are listed in Table 1. Ubiquitously expressed transcript (UXT) was selected as an internal reference gene. RT-qPCR was performed using TB Green (TaKaRa, Dalian, China). RT-qPCR data were analyzed using the 2^−ΔΔCt^ method.

### 2.7. Oil Red O Staining, Cellular TAG, and Malonyl-CoA Assay

Cells were first washed three times with pre-chilled PBS and then fixed with 10% formaldehyde solution for 30 min. Formaldehyde was discarded, PBS was washed 3 times, and Oil Red O staining solution was added and rested at room temperature for 20 min. Observation of lipid droplet deposition in cells was completed using microscope. Finally, Oil Red O was dissolved in 60% isopropanol, aliquoted into 96-well plates, and quantified at 510 nm using a colorimeter.

The triglyceride concentration of the samples was measured according to the instructions of Tissue Triglyceride (TAG) Content Assay Kit (Ap-plygen Technologies, Beijing, China). After treating the cells for 48 h, the cells were washed three times with pre-chilled PBS, and 200 μL of cell lysate was added to each well of the 6-well plate and placed on ice for 10 min. The cells were scraped into a centrifuge tube, centrifuged at 2000 r/min for 5 min, and the supernatant was removed and placed in a water bath at 70 °C for 10 min. After cooling to room temperature, the samples to be measured were mixed with the working solution and incubated at 37 °C for 15 min. The OD value was measured at 550 nm, and the standard curve was plotted to calculate the triglyceride concentration of each sample.

ELISA experiments were performed according to the instructions of Goat malonyl-CoA (MCA) ELISA Kit (MEIMIAN, Yancheng, Jiangsu, China) to detect the concentration of malonyl-CoA in each sample. First, the appropriate amount of lysate was added to the cell samples, placed on ice for 10 min, and aspirated into a centrifuge tube. Subsequently, the samples were centrifuged at 2500 r/min for 5 min, and the supernatant was aspirated as the sample to be tested. After washing 5 times with washing solution, 50 μL each of color developers A and B was added and incubated for 15 min at 37 °C. A total of 50 μL of termination solution was added after equilibration to room temperature, the OD value was measured at 450 nm, and the standard curve was plotted to calculate the malonyl-CoA concentration of each sample.

The total protein was measured with the BCA protein concentration assay kit (Boster, Wuhan, China), which was used for triglyceride and malonyl-CoA level standardization.

### 2.8. Cloning and Bioinformatics Analysis of the MCD Promoter

To obtain the MCD promoter, the primers (forward primer: ATTGGGAGTCTGGGATTAG, reverse primer: GTTCCCCAGGCAGCTGTCG) were designed after referring to the MCD-genomic DNA sequence in NCBI. Goat longissimus dorsi tissue kept in the laboratory was removed from liquid nitrogen, placed in a mortar containing liquid nitrogen, and ground to a powder. It was scraped into a centrifuge tube and treated with ultrasound to form a cell suspension. Then, the instructions of the blood/cell/tissue genomic DNA extraction kit (TIANGEN, Beijing, China) were followed to finally obtain DNA. PCR system: LA Taqase (Takara, Dalian, China) 0.2 μL, 10x Buffer 2 μL, dNTP Mix 3.2 μL, upstream and downstream primers 1 μL each, ddH_2_O 11.6 μL, and back genomic DNA template 1 μL. PCR amplification procedure: pre-denaturation at 94 °C for 4 s, denaturation at 94 °C for 10 s, annealing at 67 °C for 30 s, extension at 72 °C for 100 s, each cycle minus 1 °C, 12 cycles; denaturation at 94 °C for 10 s, annealing at 55 °C for 30 s, extension at 72 °C for 100 s, extension at 72 °C for 10 min, storage at 12 °C. The products were finally identified by sequencing. JASPARA (https://jaspar.genereg.net/) (accessed on 12 October 2022) and gene-regulation (http://gene-regulation.com/pub/programs/alibaba2/index.html) (accessed on 12 October 2022) were used to predict transcription factor binding sites of MCD promoter.

### 2.9. Statistical Analysis

All results were expressed as “mean ± SD” and analyzed based on a *t*-test in SPSS 20.0 software (IBM, Armonk, NY, USA). A difference of *p* < 0.05 is considered significant, while a difference of *p* < 0.01 is considered extremely significant.

## 3. Results

### 3.1. Characterization of MCD from Goat Liver

The goat *MCD* (*MLYCD*) gene was 1726 bp (OM937122), including 5′UTR of 27 bp, 3′UTR of 199 bp, and CDS of 1500 bp, encoding 499 amino acids. Blastn analysis showed that the sequence (CDS) similarity between goat *MCD* and *homo sapiens* (AF090834.1), *Ovis aries* (XM_004015497.5), *Bos indicus* (XM_019979237.1), and *Sus scrofa* (XM_021093741.1) was 86.33%, 99.20%, 96.93%, and 88.53%, respectively. The phylogenetic tree showed that goat *MCD* was most closely related to sheep, followed by bovine, pig, and human, and most distantly related to rat and mouse (Figure 1).

### 3.2. MCD Expression Is Negatively Correlated with Lipid Deposition

After the induction of oleic acid, cellular lipid deposition gradually increased until it started to decrease on the third day (Figure 2A). Concomitantly, *MCD* expression decreased from the first day and then continued to increase until it reached its highest value on day 6 (Figure 2B).

### 3.3. Overexpression of MCD Reduces Lipid Deposition

pcDNA3.1-MCD and pcDNA3.1 were transfected into goat intramuscular precursor adipocytes, respectively, to examine whether MCD regulates lipid deposition. Overexpression of *MCD* (657-fold increase compared to control, Figure 3A) significantly inhibited lipid droplet accumulation (Figure 3B,C). Overexpression of *MCD* also reduced the TAG and malonyl-CoA content of cells (Figure 3D,E). Furthermore, the expressions of *DGAT2* (*p* < 0.01), *ATGL* (*p* < 0.01), *FASN* (*p* < 0.01), and *ACOX1* (*p* < 0.01) were significantly up-regulated, while the expressions of *HSL* (*p* < 0.05), *CPT1A* (*p* < 0.01), and *CPT1B* (*p* < 0.01) were significantly down-regulated. *DGAT1* and *ACC* were not affected by *MCD* overexpression (Figure 3F).

### 3.4. MCD Knockdown Increases Lipid Deposition

To address the modulatory function of the MCD in the lipid metabolism of intramuscular preadipocytes, *MCD* was downregulated using siRNA-MCD molecules. RT-qPCR revealed a 73% reduction in *MCD* mRNA level (Figure 4A). Silencing *MCD* increased the accumulation of cellular lipids (Figure 4B,C) and TAG content (*p* < 0.01) compared to the control group (Figure 4D). In addition, the knockdown of *MCD* increased intracellular malonyl-CoA levels (Figure 4E). Most genes related to lipid metabolism were significantly reduced by *MCD* knockdown, including *HSL* (*p* < 0.01), *ACC* (*p* < 0.01), *FASN* (*p* < 0.01), *ACOX1* (*p* < 0.05), and *CPT1A* (*p* < 0.05), while the expression of *DGAT2* (*p* < 0.01) was increased. The expression of *DGAT1*, *ATGL*, and *CPT1B* was not affected by *MCD* silencing (Figure 4F).

### 3.5. MCD Promoter Bioinformatics Analysis

A 2025 bp goat *MCD* promoter sequence was obtained, including 1988 bp upstream of the transcription start site (TSS). Bioinformatics analysis showed that the *MCD* promoter has transcription factor binding sites, such as specificity protein 1 (SP1), sterol regulatory element binding protein 1 (SREBP1), CCAAT/enhancer binding protein α (CEBPα), peroxisome proliferator-activated receptor γ (PPARG), and Nuclear Factor Y (NF-Y). In addition, the analysis revealed the presence of GC-box and CCAT-box on the MCD promoter (Figure 5).

## 4. Discussion

MCD is an important regulator of intracellular malonyl-CoA content and fatty acid oxidation, and current research on MCD mainly focused on human diseases [12,13,19,22]. However, few studies acknowledged MCD in ruminants, especially its role in regulating intramuscular fat deposition. Similar to previous studies about mouse liver [15], our data showed a negative correlation between the expression of *MCD* and lipid deposition during the differentiation, which was then confirmed by the RNA silencing of *MCD* in goat intramuscular preadipocytes. These data were novel in that we promoted an optional target for controlling fat deposition, although different mechanisms may be involved to answer the enhancement and suppression of cellular MCD expression in goats.

To explore the mechanism by which MCD inhibits intramuscular fat deposition in goats, we next performed an expression analysis of genes related to lipid metabolism. The balance between lipid synthesis and lipolysis contributes to cellular lipid deposition. For lipid synthesis, the synthesis of new lipids begins with ACC-catalyzed carboxylation of acetyl-CoA to form malonyl-CoA, which is used as a substrate for FSNA-catalyzed production of fatty acids and finally for DGAT-catalyzed production of TAG [23]. For lipolysis, MCD catalyzes the decarboxylation of malonyl-CoA to produce acetyl-CoA, and increases in MCD activity lead to a decrease in malonyl-CoA and an increase in fatty acid oxidation [10]. Fatty triglyceride lipase (ATGL) and hormone sensitive lipase (HSL) catalyze the hydrolysis of triglycerides into free fatty acids [24,25]. The free fatty acids were then transferred to mitochondria by *CPT1*s for β-oxidation [5] or oxidated by acyl-CoA oxidase 1 (ACOX1) in the peroxisome [26]. Inhibition of *ACOX1* promotes triglyceride accumulation in mouse hepatocytes [27]. In this present study, the overexpression of *MCD* promoted cellular lipolysis by the increasement in *ATGL* (despite the slight decrease in *HSL*), which may release much more free fatty acids [15]. The increased expression of *ACOX1* and decreased expression of *CPT1s* may illustrate that it was the peroxisome, but not the mitochondria, that responded to the increased release of exact free fatty acids induced by *MCD* overexpression. Meanwhile, much more free fatty acids may also provide ligands for the regulators of lipid synthesis [28]. These regulators, such as SP1, SREBP1, C/EBPα, and PPARG [29,30], may promote the expression of *MCD* by directly binding to the promoter according to the promoter analysis results. This may explain the increasement in *FASN* and *DGAT2*. However, the lower concentration of malonyl-CoA (the substrate for lipid synthesis) by *MCD* overexpression hampers lipid synthesis [31] despite the increasement in lipogenesis-related genes. These data may ultimately lead to the suppression of lipid accumulation.

We also observed that consistent with the increased cellular lipid accumulation, the silencing of *MCD* activated the lipogenesis (increased expression of *DGAT2* [32]) and suppressed the lipolysis (decreased expression of *ATGL* and *HSL*), resulting in the lower concentration of cellular free fatty acids and downregulated expression of fatty acid synthesis by *FASN* and *ACC*. Meanwhile, the increased concentration of malonyl-CoA may explain the inhibition of fatty acid oxidation mediated by CPT1s [33]. Although we have no idea about the mechanism underlying the increased expression of *DGAT2*, these data underlined the positive role of lipid deposition induced by *MCD* suppression in goats.

## 5. Conclusions

In conclusion, we cloned 1726 bp of the CDS sequence and 2025 bp of the promoter sequence of the goat *MCD* gene. The expression alteration of MCD may negatively regulate cellular lipid deposition, with the involvement of C/EBPα, SP1, SREBP1, and PPARG. Understanding the regulatory role of MCD in intramuscular fat deposition in goats may help improve goat meat quality.

## Figures and Tables

**Figure 1 genes-14-00440-f001:**
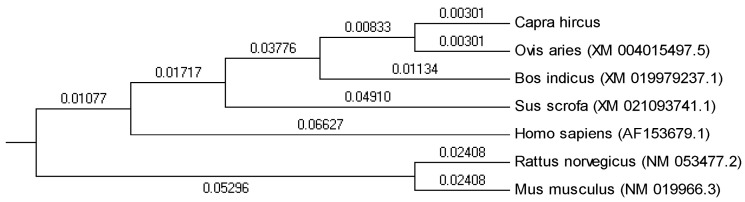
Phylogenetic tree based on *MCD* gene sequences of 7 representative animals constructed by MEGA 5 software. The number on the branch line represents the evolutionary branch length.

**Figure 2 genes-14-00440-f002:**
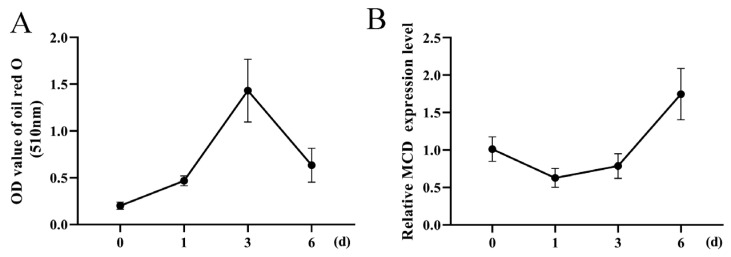
Relative expression level of *MCD* during intramuscular adipocyte differentiation. (**A**) Lipid deposition during differentiation of goat intramuscular preadipocytes. (**B**) Relative expression of *MCD* in intramuscular preadipocytes at day 0, 1, 3, 6 of induced differentiation.

**Figure 3 genes-14-00440-f003:**
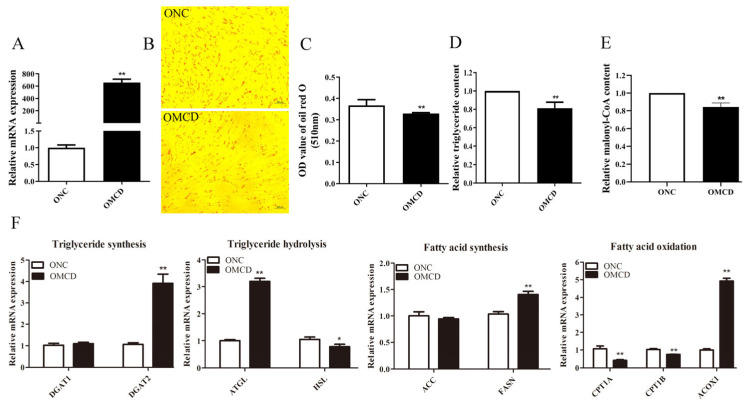
Overexpression of *MCD* inhibits fat droplet accumulation and triacylglycerol (TAG) synthesis in goat intramuscular preadipocytes. (**A**) *MCD* overexpression efficiency assay. (**B**) Cellular lipid droplets were observed through a microscope (100 μm) after using Oil Red O staining. (**C**) The OD assay was performed at 510 nm by dissolving Oil Red O stain with isopropyl alcohol. (**D**) Effect of overexpression of *MCD* on intracellular TAG content. (**E**) Detection of intracellular malonyl-CoA levels by ELISA. (**F**) Effect of overexpression of *MCD* on mRNA levels of genes related to lipid metabolism. After transfection of pcDNA3.1 or pcDNA.3.1-MCD into cells for 48 h, RT-qPCR assays were performed. Data are presented as “Means ± SD”, *n* = 3. ** *p* < 0.01, * *p* < 0.05. ONC: pcDNA3.1 treatment as control; OMCD: pcDNA3.1-MCD treatment.

**Figure 4 genes-14-00440-f004:**
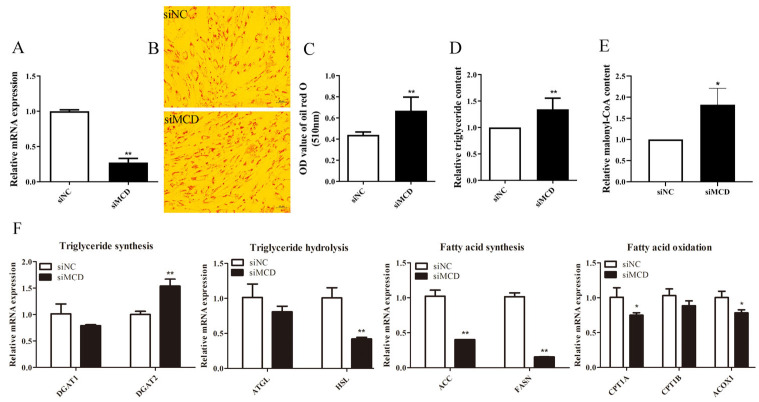
Silencing of *MCD* increased cellular lipid accumulation and triacylglycerol (TAG) content. (**A**) *MCD* interference efficiency detection. (**B**) Cellular lipid droplets were observed through a microscope (10 μm) after using Oil Red O staining. (**C**) The OD assay was performed at 510 nm by dissolving Oil Red O stain with isopropyl alcohol. (**D**) Effect of knockdown of *MCD* on intracellular TAG content. (**E**) Detection of intracellular malonyl-CoA levels by ELISA. (**F**) Effect of *MCD* knockdown on expression of genes related to lipid metabolism. Results are presented as “Means ± SD”, *n* = 3. ** *p* < 0.01, * *p* < 0.05. siNC: Negative control siRNA treatment as control; siMCD: siRNA-MCD treatment.

**Figure 5 genes-14-00440-f005:**
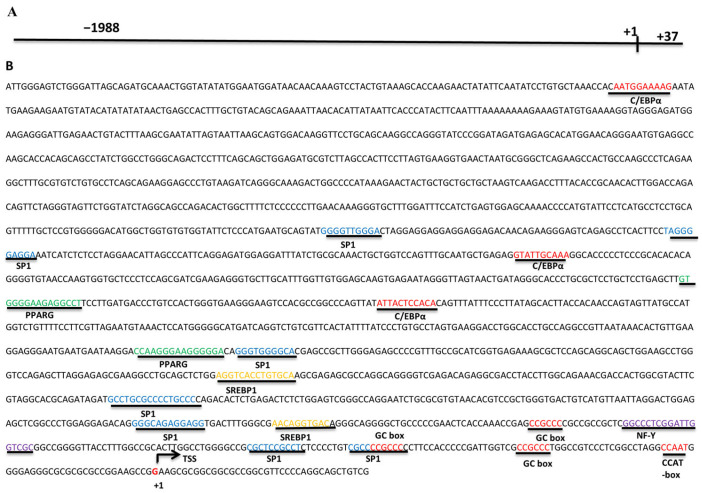
Sequence identification of the goat *MCD* promoter.(**A**) Length of the goat MCD promoter sequence obtained by cloning. (**B**) Analysis of regulatory elements in the goat MCD promoter sequence.

**Table 1 genes-14-00440-t001:** Primer information for RT-qPCR.

Gene Name	Accession Numbers	Primer Sequence (5′-3′)	Tm (°C)
*MCD*	OM937122	Forward: AAGTGCTCCAGAGAATCAGCG Reverse: GGCTCAGCGAGTAGAAGATGG	60
*DGAT1*	NM_174693	Forward: CCACTGGGACCTGAGGTGTC	60
		Reverse: GCATCACCACACACCAATTCA	
*DGAT2*	BT030532.1	Forward: CATGTACACATTCTGCACCGATT Reverse: TGACCTCCTGCCACCTTTCT	60
*ATGL*	GQ918145	Forward: GGAGCTTATCCAGGCCAATG Reverse: TGCGGGCAGATGTCACTCT	60
*HSL*	EU273879	Forward: GGGAGCACTACAAACGCAACG Reverse: TGAATGATCCGCTCAAACTCG	60
*ACC*	NM_174224.2	Forward: CTCCAACCTCAACCACTACGG Reverse: GGGGAATCACAGAAGCAGCC	60
*FASN*	DQ915966.3	Forward: GGGCTCCACCACCGTGTTCCA Reverse: GCTCTGCTGGGCCTGCAGCTG	60
*CPT1A*	XM_018043311.1	Forward: TGACGGCTCTGGCACAAGAT Reverse: CGCGAAGTAGTTGCTATTCAC	60
*CPTAB*	MH340532	Forward: ACGAGGAGTCTCACCACTACG Reverse: GTGTGAAGGACTTGTCGAACCA	60
*ACOX1*	NM_00103528	Forward: CGAGTTCATTCTCAACAGTCCT Reverse: GCATCTTCAAGTAGCCATTATCC	60
*UXT*	XP_005700899.1	Forward: GCAAGTGGATTTGGGCTGTAAC Reverse: ATGGAGTCCTTGGTGAGGTTGT	60

Abbreviations: DGAT1, diacylglycerol acyltransferas 1; DGAT2, diacylglycerol acyltransferase 2; ATGL, adipose triglyceride lipase; HSL, hormone-sensitive lipase; ACC: acetyl coa carboxylase; FASN, fatty acid synthase; CPT1A, carnitine palmitoyltransferase 1A; CPT1B, carnitine palmitoyltransferase 1B; ACOX1: acyl-CoA oxidase 1; UXT, ubiquitously expressed transcript.

## Data Availability

Not applicable.

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
