# Peer review of "MCD Inhibits Lipid Deposition in Goat Intramuscular Preadipocytes"

_genes, 2023, doi:10.3390/genes14020440_

Round 1

Reviewer 1 Report

Authors in their article titled "Malonyl-CoA decarboxylase inhibits lipid deposition in goat  intramuscular preadipocytes" cloned the cDNA and promoter region of Malonyl-CoA decarboxylase and provide evidence about its expression. Author report novel and interesting information. There are some missing gabs that I feel that authors could cover to improve the quality of their text. Therefore I suggest a major revision according to my comments below to improve the quality of the text.

Introduction

Authors could refer to differences between ruminants  and other specie like human in regard to lipogenesis (briefly) and also to highlight any difference in the biochemistry pathway where malonyl CoA decarboxylase participates. They could also clear which are the differences of respective enzymes in cytocolic, mitochondria or peroxisomes. Are these enzymes encoded by the same genes. Authors should also specify which gene they cloned (i.e. that of encoding cytocolic MCD?)

Materials and methods

There are some missing information in some procedures. So please refer to my comments following line by line

l 95 each cycle minus 1 °C for how many cycles?

l121 specify the initial amount of cells used

l 124 The same as above, specify the initial quantity of RNA used

161 you refer to a 96-plate but how this is connected with the method describe. you have not described anything

l. 172 how dna was isolated? please describe briefly

l 173-175 please refer to  the parameters used in the programs for promoter characterization.

l.177 which analysis is based on t-test what do you compare

Results

l. 183 Please  include in  materials and methods about blast analysis and the construction of phylogenetic tree. In addition specify in Fig 1 what the numbers in branches represent

l 222-232 authors refer to knock down results but they have not describe the procedure and at which site was made the knock-down

l. 243-248 Author should refer to the existence or not of TATA  box, CAAT box or similar elements that are considered as core promoter elements. In addition it would be useful to determine any potential transcription start site

Discussion

It would be useful to make a comparison with human or rat/mouse according to your results related to expression.  In addition be careful with regulating factors because these are considered as potential (predicted) by the bioinformatic analysis. Finally you may propose a potential model  how MCD influence IMF.

Minor points

l. 16 omit by

l20 amino acids (omit residues)

l37 higher than what?

l 48 and is highly expressed

 l 102 -104 better phrasing. Do you describe the design?

 l 105 there is no underlined position

l 106 MCD broth please explain what do you mean?

l287-290 please rephrase for better meaning

Author Response

Comments and Suggestions for Authors

Authors in their article titled "Malonyl-CoA decarboxylase inhibits lipid deposition in goat  intramuscular preadipocytes" cloned the cDNA and promoter region of Malonyl-CoA decarboxylase and provide evidence about its expression. Author report novel and interesting information. There are some missing gabs that I feel that authors could cover to improve the quality of their text. Therefore I suggest a major revision according to my comments below to improve the quality of the text.

Introduction

Authors could refer to differences between ruminants  and other specie like human in regard to lipogenesis (briefly) and also to highlight any difference in the biochemistry pathway where malonyl CoA decarboxylase participates. They could also clear which are the differences of respective enzymes in cytocolic, mitochondria or peroxisomes. Are these enzymes encoded by the same genes. Authors should also specify which gene they cloned (i.e. that of encoding cytocolic MCD?)

AU: Thank you very much for your suggestion. This is a very good suggestion. However, lipid differences between ruminants and non-ruminants are a very complex scientific issue. These may require future in-depth studies to understand a specific lipid metabolism difference between ruminants and other species. In addition, the role played by MCD in different localizations needs to be studied in the future under the same conditions. And for specific genetic clones, in the future we will note.

Materials and methods

There are some missing information in some procedures. So please refer to my comments following line by line

l 95 each cycle minus 1 °C for how many cycles?

AU: In this study, we used touchdown PCR with one degree drop per cycle, and the annealing temperature from 68 degrees to 55 degrees should be 13 cycles. We have made a correction in the text.

l121 specify the initial amount of cells used

AU: Thank you for your reminder. This initial cell volume assay that you mentioned is a good suggestion. Since there were no experiments involving cell proliferation or apoptosis in this study, the specific initial cell inoculum was not tested. For the RT-qPCR experiments, we finally used 1ug of RNA for reverse transcription. For oil red O staining, malonyl-CoA and TAG assays, we corrected the results by BCA measurement of protein. Once again, thank you for your suggestions and we will do a better job in this regard in the future.

l 124 The same as above, specify the initial quantity of RNA used

AU: Thank you for your reminder. We performed reverse transcription experiments using 1 µg of RNA

161 you refer to a 96-plate but how this is connected with the method describe. you have not described anything.

AU: 96-well plates were used to dispense dissolved Oil Red O and measured at 510 nm, and we have reorganized the language as follows: “Finally, Oil Red O was dissolved in 60% isopropanol, aliquoted into 96-well plates and quantified at 510 nm using a colorimeter.”

  1. 172 how dna was isolated? please describe briefly

AU: Thank you for your reminder. We have added the DNA isolation process in the paper as follows: “Goat longissimus dorsi tissue kept in the laboratory was removed from liquid nitrogen, placed in a mortar containing liquid nitrogen, and ground to a powder. It was scraped into a centrifuge tube and treated with ultrasound to form a cell suspension. Then follow the instructions of the blood/cell/tissue genomic DNA extraction kit (TIANGEN, Beijing, China) to finally obtain DNA.”

l 173-175 please refer to the parameters used in the programs for promoter characterization.

AU: Thanks to the reviewer for the reminder. We have placed the online software URL for analyzing promoters in the Materials and Methods section.

l.177 which analysis is based on t-test what do you compare

AU: Thank you for your review. We illustrate the data for t-test comparisons in the text. As follows: “All results were expressed as "mean ± SD" and analyzed based on t-test in SPSS 20.0 software (IBM, USA).”

Results

  1. 183 Please include in materials and methods about blast analysis and the construction of phylogenetic tree. In addition specify in Fig 1 what the numbers in branches represent

AU: We have added method descriptions for blast analysis and the construction of phylogenetic tree to the Materials and Methods section.

l 222-232 authors refer to knock down results but they have not describe the procedure and at which site was made the knock-down

AU: This study interfered with the MCD gene by siRNA and the corresponding siRNA sequences are in the Materials Methods section. The knockout technique was not used in this study, so the process and loci of knockout were not described.

  1. 243-248 Author should refer to the existence or not of TATA box, CAAT box or similar elements that are considered as core promoter elements. In addition it would be useful to determine any potential transcription start site

AU: Thanks to the reviewer's suggestion, we analyzed the MCD promoter sequence again and found the presence of GC-box and CCAT-box on the MCD promoter.

Discussion

It would be useful to make a comparison with human or rat/mouse according to your results related to expression. In addition be careful with regulating factors because these are considered as potential (predicted) by the bioinformatic analysis. Finally you may propose a potential model  how MCD influence IMF.

AU: Thank you for your suggestion. We compare the differences in lipid accumulation due to changes in MCD expression in goats and mice in the first paragraph of the Discussion section. As follows: “Similar with previous studies in mouse liver, our data showed a negative correlation between the expression of MCD and lipid deposition during the differentiation, which was then confirmed by RNA silencing of MCD in goat intramuscular preadipocytes.” The role of MCD on lipid metabolism among different species may need further study in the future, especially in ruminants thought to be unknown except for goats.

Minor points

  1. 16 omit by

AU: Thank you for your reminder. We removed the by.

l20 amino acids (omit residues)

AU: Thank you for your reminder. We have removed residues in the text

l37 higher than what?

AU: Thank you for your reminder. We have made additions to the text as follows: “Thus, the quality of goat meat is in higher demand.”

l 48 and is highly expressed

AU: Thank you for your reminder. We have made changes in the text, as follows: “Malonyl-CoA decarboxylase, which catalyze the decarboxylation of malonyl-CoA, was first characterized from the goose uropygial gland and is highly expressed in rat liver, heart, and adipose tissue.”

l 102 -104 better phrasing. Do you describe the design?

AU: Thank you for your suggestion. We have made changes in the text.

l 105 there is no underlined position

AU: Thank you for your reminder. We have added underscores to the corresponding places of restriction endonucleases in the text.

l 106 MCD broth please explain what do you mean?

AU: Thank you for your reminder. We have modified the error as follows: “PCR amplification was performed using pMD19-T-MCD plasmid as template. PCR products were double digested, purified and ligated with pcDNA3.1 vector, and finally sequenced.”

l287-290 please rephrase for better meaning

AU: Thank you for your suggestion.

Reviewer 2 Report

Title

Malonyl-CoA decarboxylase inhibits lipid deposition in goat 2 intramuscular preadipocytes

The title should include any word coincide with scope of the journal (genes)

Introduction

In the present study, we elucidate the function of MCD in regulating cellular lipid metabolism using the treatment of 63 MCD knockdown and overexpression in goats

The sentence should be in the past

Materials and Methods  Animals, cell isolation and culture

Ethical approval statement should be included

Author Response

Comments and Suggestions for Authors

Title

Malonyl-CoA decarboxylase inhibits lipid deposition in goat 2 intramuscular preadipocytes

The title should include any word coincide with scope of the journal (genes)

AU: Thank you for your reminder. We changed the title to “MCD inhibits lipid deposition in goat intramuscular preadipocytes”.

Introduction

In the present study, we elucidate the function of MCD in regulating cellular lipid metabolism using the treatment of 63 MCD knockdown and overexpression in goats The sentence should be in the past

AU: Thank you for your reminder. We have made changes in the text, as follows: “In the present study, we elucidated the function of MCD in regulating cellular lipid metabolism using the treatment of MCD knockdown and overexpression in goats.”

Materials and Methods  Animals, cell isolation and culture

Ethical approval statement should be included

AU: Thank you for your reminder. We have added an ethics statement to the "Animals, cell isolation and culture" section. This is as follows: “The animal studies were approved by the Southwest Minzu University Animal Care and Use Committee.”

Round 2

Reviewer 1 Report

Authors incorporate in their manuscript most of my comments improving the quality of the text. Therefore, I recommend the further publication of the article.